# Cadmium Complexed with β2-Microglubulin, Albumin and Lipocalin-2 rather than Metallothionein Cause Megalin:Cubilin Dependent Toxicity of the Renal Proximal Tubule

**DOI:** 10.3390/ijms20102379

**Published:** 2019-05-14

**Authors:** Johannes Fels, Bettina Scharner, Ralf Zarbock, Itzel Pamela Zavala Guevara, Wing-Kee Lee, Olivier C. Barbier, Frank Thévenod

**Affiliations:** 1Department of Physiology, Pathophysiology & Toxicology and ZBAF (Centre for Biomedical Education and Research), Faculty of Health, School of Medicine, Witten/Herdecke University, D-58453 Witten, Germany; Johannes.Fels@uni-wh.de (J.F.); Bettina.Scharner@uni-wh.de (B.S.); Ralf.Zarbock@uni-wh.de (R.Z.); Wing-Kee.Lee@uni-wh.de (W.-K.L.); 2Departamento de Toxicología, Centro de Investigación y de Estudios Avanzados del Instituto Politécnico Nacional (CINVESTAV-IPN), Mexico 07360, México; pamela0191@hotmail.com (I.P.Z.G.); obarbier@cinvestav.mx (O.C.B.)

**Keywords:** Cadmium nephrotoxicity, proximal tubule, megalin, cubilin, metallothionein, albumin, transferrin, β2-microglobulin, lipocalin-2

## Abstract

Cadmium (Cd^2+^) in the environment is a significant health hazard. Chronic low Cd^2+^ exposure mainly results from food and tobacco smoking and causes kidney damage, predominantly in the proximal tubule. Blood Cd^2+^ binds to thiol-containing high (e.g., albumin, transferrin) and low molecular weight proteins (e.g., the high-affinity metal-binding protein metallothionein, β2-microglobulin, α1-microglobulin and lipocalin-2). These plasma proteins reach the glomerular filtrate and are endocytosed at the proximal tubule via the multiligand receptor complex megalin:cubilin. The current dogma of chronic Cd^2+^ nephrotoxicity claims that Cd^2+^-metallothionein endocytosed via megalin:cubilin causes renal damage. However, a thorough study of the literature strongly argues for revision of this model for various reasons, mainly: (i) It relied on studies with unusually high Cd^2+^-metallothionein concentrations; (ii) the *K_D_* of megalin for metallothionein is ~10^5^-times higher than (Cd^2+^)-metallothionein plasma concentrations. Here we investigated the uptake and toxicity of ultrafiltrated Cd^2+^-binding protein ligands that are endocytosed via megalin:cubilin in the proximal tubule. Metallothionein, β2-microglobulin, α1-microglobulin, lipocalin-2, albumin and transferrin were investigated, both as apo- and Cd^2+^-protein complexes, in a rat proximal tubule cell line (WKPT-0293 Cl.2) expressing megalin:cubilin at low passage, but is lost at high passage. Uptake was determined by fluorescence microscopy and toxicity by MTT cell viability assay. Apo-proteins in low and high passage cells as well as Cd^2+^-protein complexes in megalin:cubilin deficient high passage cells did not affect cell viability. The data prove Cd^2+^-metallothionein is not toxic, even at >100-fold physiological metallothionein concentrations in the primary filtrate. Rather, Cd^2+^-β2-microglobulin, Cd^2+^-albumin and Cd^2+^-lipocalin-2 at concentrations present in the primary filtrate are taken up by low passage proximal tubule cells and cause toxicity. They are therefore likely candidates of Cd^2+^-protein complexes damaging the proximal tubule via megalin:cubilin at concentrations found in the ultrafiltrate.

## 1. Introduction

Cadmium (Cd^2+^) pollution is increasing in the environment worldwide due to industrial activities and because it cannot be further degraded. Compared to other transition metals, Cd^2+^ and Cd^2+^ compounds are relatively water soluble, more bio-available, and therefore tend to bio-accumulate [1]. Chronic Cd^2+^ exposure, mainly from contaminated dietary sources and cigarette smoking [2], leads to Cd^2+^ accumulation in the human body, especially in the kidneys where renal tubular damage is probably the most critical health effect.

Following absorption in the intestine and/or lungs, Cd^2+^ in the blood initially binds largely to albumin and other thiol-containing high-molecular (HMWP) and low-molecular weight proteins (LMWP) in the plasma, including metallothioneins (MT), as well as to blood cells [2]. MT, a cysteine-rich metal-binding protein, binds both physiological Zn^2+^ ions and toxic Cd^2+^ ions through the thiol group of its cysteine residues with very high affinity (*K_D_* for Cd^2+^ ~10^−14^ mol/L) [3].

In the literature, a presumed model prevails wherein blood Cd^2+^ is taken up by hepatocytes [4] or Kupffer cells [5] in which free cytosolic Cd^2+^ is thought to induce synthesis of MT to bind and detoxify Cd^2+^. Supposedly, Cd^2+^-MT is steadily released into the bloodstream as the cells undergo necrosis, either through normal cell turnover or as a result of Cd^2+^ injury, and redistributes to the kidney. Since intravenously injected Cd^2+^-MT in mice is quickly cleared from the plasma by the kidneys [6], this protein fraction in the circulation is thought to be of great importance for Cd^2+^ shuttling from the liver to the kidney during long-term exposure [7,8,9]. Although some experimental evidence exists from animal studies that Cd^2+^ redistributes from the liver to the kidney during chronic exposure to high Cd^2+^ concentrations, this is not the case for low environmentally relevant blood concentrations [9], which in humans range between 0.03 and 0.5 µg/L (~0.3–5 nmol/L) (reviewed in reference [10]). In fact, the plasma Cd^2+^-MT concentrations following intravenous injections exceeded physiological MT concentrations by >2000-fold [11,12]. With a molecular mass (MM) of ~3.5–14 kDa, (Cd^2+^)-MT easily crosses the glomerular barrier (cut-off ~80 kDa) and is thought to be completely endocytosed from the ultrafiltrate in the S1-segment of the proximal tubule (PT) [13,14], by the multiligand, endocytic-membrane receptor complex megalin:cubilin [15]. Endocytosed Cd^2+^-MT is trafficked to acidic late endosomes and lysosomes where the protein moiety is degraded and Cd^2+^ is extruded via divalent metal transporter 1 (DMT1) into the cytosol to develop toxicity [16].

Megalin:cubilin also binds and retrieves other filtered proteins, such as the LMWP β2-microglobulin (β2M; MM ~11 kDa) neutrophil gelatinase-associated lipocalin (NGAL)/lipocalin-2 (Lcn2; MM ~22–25 kDa), and α1-microglobulin (α1M; MM ~27 kDa), as well as the HMWP albumin (Alb; MM ~67 kDa) and transferrin (Tf; MM ~76 kDa) (reviewed in reference [15]). In contrast to LMWP, which are freely filtered and are present in similar concentrations in plasma and ultrafiltrate, only lower concentrations of HMWP will be found in the ultrafiltrate based on their estimated glomerular sieving coefficients (reviewed in reference [17]). Furthermore, the aforementioned proteins also bind divalent metal ions, including Cd^2+^, with *K_D_* values of ~10^−6^ mol/L and maximally two binding sites [18,19,20].

Hence, megalin:cubilin represents a high-capacity receptor for endocytosis that prevents protein loss from the body into the urine [21]. Although the affinity of megalin:cubilin to most of its protein ligands is high (*K_D_* varies between 20–600 nmol/L) (reviewed in reference [17]), its affinity to MT is in the high micromolar range [22,23], which practically excludes MT uptake by the PT, taking into account that plasma (and ultrafiltrate) concentrations of MT measured are in the high picomolar to low nanomolar range [11,12]. This has been overlooked in the literature so far and indicates that Cd^2+^-MT is not the primary source of complexed Cd^2+^ responsible for damaging the kidney PT.

In the present study we have determined the uptake and toxicity of Cd^2+^ complexed to LMWP β2M, α1M and Lcn2 and the HMWP Alb and Tf using a rat PT cell line (WKPT-0293 Cl.2) expressing megalin:cubilin. The data reiterate that Cd^2+^-MT is not toxic, even at concentrations that exceed the physiological MT concentrations in the ultrafiltrate by ~100-fold. Rather, Cd^2+^-β2M, Cd^2+^-Alb and Cd^2+^-Lcn2 are taken up by PT cells and cause toxicity at ultrafiltrate concentrations and therefore represent more likely candidates of Cd^2+^-protein complexes that damage the renal PT via megalin:cubilin-dependent endocytosis.

## 2. Results

The WKPT-0293 Cl.2 cell line, derived from the S1-segment of rat PT [24], is one of few renal PT cell lines to express megalin and cubilin (Figure 1) [23,25,26], which is lost at passage numbers exceeding 40. Real-time PCR confirmed loss of megalin mRNA in high passage (*p* > 100) compared to low passage (*p* ≤ 40 cells) (Figure 1A). Importantly, low passage cells also express higher cubilin (*Cubn*) mRNA levels than high passage cells (Figure 1A) since endocytosis of Tf, Alb and α1M requires cubilin binding and interaction with megalin to occur [27,28,29] (reviewed in reference [15]). In contrast, endocytosis of MT, Lcn2 and β2M is exclusively megalin-dependent. Interestingly, mRNA expression levels of megalin were higher than cubilin in low passage cells (Figure 1A), which is in agreement with their relative protein expression in rat PT [21]. Immunofluorescence staining using a commercial monoclonal antibody against whole megalin whose epitope has not yet been characterized (courtesy of Novus Biologicals) clearly shows staining in permeabilized low passage, but not in high passage cells (Figure 1B,C). No positive staining was detected in non-permeabilized low passage cells, which indicates that the antibody is directed against an intracellular epitope. Immunoblotting of cubilin in low and high passage cells (Figure 1D) confirmed cubilin protein expression in low, but not in high passage cells.

MT has been demonstrated to be a low affinity ligand of megalin in previous studies by surface plasmon resonance [22] or uptake of Alexa Fluor 488-coupled MT in WKPT-0293 Cl.2 cells [23]. As shown in Figure 2A, Cd^2+^-MT toxicity increased as a function of concentration in low passage cells but became only significantly different to high passage cells at the highest tested concentration of 10 µmol/L Cd^2+^-MT (≅ 70 µmol/L Cd^2+^) after 24 h exposure. In contrast, high passage cells did not show significant toxicity at all concentrations tested, further demonstrating the importance of megalin for Cd^2+^-MT endocytosis. Similarly, a high concentration of 7.14 µmol/L Cd^2+^-MT (≅ 50 µmol/L Cd^2+^), which was used in WKPT-0293 Cl.2 cells in previous studies [23], showed 59 ± 4 % cell viability (*n* = 5; *p* < 0.01) in low passage cells expressing megalin and 98 ± 13% cell viability in megalin-deficient high passage cells (*n* = 4; *p* < 0.01) whereas no difference was observed for MT alone (Figure 2B). In contrast, 50 µmol/L CdCl_2_ (24 h) decreased cell viability to 0% (*n* = 4) in both high and low passage cells, indicating that the mechanisms of CdCl_2_ toxicity differ from Cd^2+^-MT toxicity and also proving that the totality of excess free Cd^2+^ has been removed from Cd^2+^-MT (see Section 4.2).

Since in vivo relevant concentrations of Cd^2+^-MT in plasma [11,12] and primary filtrate, which are in the low nanomolar range (Table 1), are unlikely to cause PT toxicity because megalin:cubilin affinity for MT is too low, alternative candidate Cd^2+^-protein ligands for megalin:cubilin were investigated, i.e., β2M, Lcn2, Tf, Alb and α1M [15,17]. Similarly to MT, the concentration of the cubilin ligand Tf in the primary filtrate is in the low nanomolar range at 2 nmol/L and binds megalin with high affinity (Table 1). In contrast, β2M, Alb and α1M are present in the primary filtrate at ~100, 50 and 92 nmol/L, respectively (Table 1). Whereas β2M and Alb have known *K_D_* for megalin and cubilin, respectively, at high nanomolar concentrations (Table 1), the *K_D_* of α1M for cubilin is unknown [15]. Finally, Lcn2 has a higher expected primary filtrate concentration of ~650 nmol/L and a *K_D_* for megalin binding that is similar to Tf (see Table 1).

To mimic the Cd^2+^-protein concentrations measured in the filtrate (to which PT cells are exposed to), low passage cells were also incubated for 24 h with 100 nmol/L of β2M, and Alb or α1M as apo-proteins or as Cd^2+^ complexes, and MTT absorbance was measured (Figure 3A). MT at the same concentration was included for comparison. Similarly as shown in Figure 2A, Cd^2+^-MT was not toxic (although its filtrate concentration is ~100-fold lower). In contrast, Cd^2+^-β2M significantly decreased cell viability (59 ± 6%; *n* = 4; Figure 3A). A concentration of Tf at the approximately expected filtrate concentration (5 nmol/L Cd^2+^-Tf with 10 nmol/L Cd^2+^ assuming 2 Cd^2+^-binding sites [18]) did not reduce cell viability of low passage WKPT-0293 Cl.2 cells. Even at 100 nmol/L Cd^2+^-Tf showed no significant decrease of cell viability (90 ± 13% cell viability; *n* = 8), indicating that it is not likely to cause toxicity in vivo. Interestingly, Cd^2+^-Alb (assuming 1 Cd^2+^-binding site [19]) significantly decreased cell viability (59 ± 8%; *n* = 7; Figure 3A). Yet 100 nmol/L Cd^2+^-α1M did not induce significant toxicity in low passage cells (82 ± 14% cell viability; *n* = 9). At 100 nmol/L, Cd^2+^-Lcn2 was not toxic (103 ± 7% of cell viability; *n* = 5; Figure 3B). However, Cd^2+^-Lcn2 at 1 µmol/L, a concentration close to that in the primary filtrate (Table 1) also showed significant toxicity in low passage cells (64 ± 4%; *n* = 6; Figure 3B).Viability of high passage cells was unaffected by Cd^2+^-β2M, Cd^2+^-Alb or Cd^2+^-Lcn2, when tested under the same conditions.

Alb uptake by WKPT-0293 Cl.2 cells was confirmed by fluorescence microscopy of FITC-coupled Alb (100 nmol/L for 24 h). Interestingly, although low passage cells showed increased internalization of FITC-Alb (Figure 4), the difference to high passage cells was not as pronounced as megalin:cubilin expression (Figure 1) and toxicity in low passage cells (Figure 3A). This was also the case for β2M uptake.

## 3. Discussion

The current dogma of chronic Cd^2+^ nephrotoxicity assumes that (1) Cd^2+^ exposure results in initial accumulation of the metal ion in the liver, where it induces MT synthesis and binds to the protein; (2) as a result of normal cell turnover, or hepatic damage some of the Cd^2+^-MT is released into the blood; (3) Cd^2+^-MT in the circulation is rapidly cleared by the renal glomeruli and reabsorbed by the PT; (4) this translocation eventually leads to nephrotoxicity [4]. Thus MT is claimed to serve both as a storage and a transport protein for Cd^2+^ [33].

Yet, a thorough assessment of the literature clearly indicates that this model can no longer be accepted. Firstly, and most importantly, susceptibility of MT-null mice to chronic CdCl_2_-induced nephrotoxicity indicates that renal PT injury is not mediated by the Cd^2+^-MT complex [34]. Secondly, renal uptake of ultrafiltrated Cd^2+^-MT was only observed with very high concentrations of parenterally applied acute Cd^2+^-MT injections varying between 0.7–2 µmol/L [35,36,37,38,39] and is an inappropriate model for the study of chronic Cd^2+^-induced nephrotoxicity [40]. Finally, the in vivo evidence for redistribution of liver Cd^2+^-MT to the kidney is based on experimental models that rely on excessive hepatic Cd^2+^ accumulation or exposure to hepatotoxic agents [7,41,42,43,44] and is not observed in mice subjected to chronic administration of Cd^2+^ via drinking water, which mimics chronic exposure from dietary sources [9].

Overall, in light of our reasoning and in context of a *K_D_* of megalin:cubilin to MT in the high micromolar range [22,23], it was reasonable to assume that other filtered Cd^2+^-protein complexes may at least partly mediate chronic PT toxicity. Complex formation of Alb, β2M and Tf with Cd^2+^ and other divalent metal ions with a *K_D_* of around 1 µmol/L and maximally two Cd^2+^ binding sites has been described [18,19,20]. We also investigated Lcn2 and α1M that are both filtered and ligands of megalin [31] and cubilin [29], respectively, for which no binding data to Cd^2+^ are available in the literature. Cd^2+^-β2M, Cd^2+^-Alb and Cd^2+^-Lcn2 at concentrations that are compatible with the ultrafiltrated concentrations of these proteins in vivo caused toxicity of low passage WKPT-0293 Cl.2 cells expressing megalin:cubilin and were internalized by the cells. It may be debated that these proteins exhibit relatively low affinities to Cd^2+^ compared to MT, indicating that at steady-state they will be maximally saturated by ~1% to form complexes with blood Cd^2+^ (with a concentration of 0.3–5 nM [10]) whereas MT in the circulation will be Cd^2+^-saturated (based on equivalent low nanomolar concentrations of MT and Cd^2+^ and a *K_D_* for Cd^2+^ ~10^−14^ mol/L [3]). Yet the relatively high concentrations of β2M, Alb and Lcn2 in the primary filtrate compared to MT, their high binding affinity to megalin (Table 1) and the multiplicative effect of continuous glomerular filtration makes them more likely to contribute to chronic renal PT toxicity than Cd^2+^-MT whose concentration in the primary filtrate is at least 10^5^-times lower [11,12] than its affinity to megalin [22]. Of note, β2M is known as an early urinary biomarker of renal PT toxicity induced by chronic low Cd^2+^ exposure, because when its reabsorption via megalin:cubilin is disrupted [38,45,46] (reviewed in reference [47]), it is consequently excreted into the urine.

Strikingly, the difference in uptake of Alb and β2M between low and high passage cells was less pronounced than expected from the megalin:cubilin expression (Figure 1) and toxicity data, suggesting that other entry pathways for these (Cd^2+^) protein complexes could contribute to endocytosis without leading to toxicity, such as transcytosis [48].

From the current data Cd^2+^-MT is unlikely to be endocytosed in the PT, and because only 0.02–03% of filtered proteins, including MT, are excreted with the urine, [49] more distal nephron segments must be involved in MT retrieval. Indeed, the renal medulla accumulates significant amounts of both Cd^2+^ and Cd^2+^-MT in humans, and concentrations of both Cd^2+^ compounds can reach ~50% of the levels found in the cortex [50,51]. MT was detected by immunohistochemistry in distal nephron segments of rodent and human kidney [52,53,54], and the expression of MT was induced by exposure to Cd^2+^ [53,54]. A high-affinity receptor for MT has been identified in the distal nephron, the Lcn2 receptor (Lcn2-R/SLC22a17/BOCT [brain organic cation transporter]). Lcn2-R is expressed apically in the distal convoluted tubule (DCT) and collecting duct (CD), mainly inner medullary [55]. Chinese hamster ovary cells transiently expressing Lcn2-R and cultured mouse DCT cell line endogenously expressing Lcn2-R internalized sub-micromolar concentrations of fluorescence-labelled MT (*K_D_* ~100 nmol/L), Tf or Alb whose uptake was prevented by picomolar Lcn2 concentrations, which indicated that the Lcn2-R mediates uptake of these proteins [55]. Exposure of both cell lines expressing Lcn2-R to 700 nM Cd^2+^-MT induced cell death that could be reduced by Lcn2 [55], which indicates that Cd^2+^-MT is a high-affinity ligand of Lcn2-R that mediates endocytosis and toxicity of Cd^2+^-MT in the distal nephron. Hence, although Cd^2+^-MT may not be directly involved in Cd^2+^-induced PT damage, it is likely responsible for renal DCT/CD/medulla Cd^2+^ uptake and toxicity. Considering that up to 90% of the primary renal fluid is reabsorbed by the PT and Henle loop and about 1% is excreted as urine (reviewed in references [56,57]), luminal Cd^2+^-MT concentrations in the distal nephron may increase up to 500 nmol/L (cf. [11,12]), which is in the range of the binding affinity of the Lcn2-R for MT [55]. Investigation of distal nephron damage has been neglected so far, and only rare evidence for chronic Cd^2+^ toxicity of the distal portions of the nephron has been obtained, both in experimental animals [58,59] and Cd^2+^-exposed workers [60].

In summary, at relevant concentrations present in the primary filtrate, renal PT cells do not succumb to Cd^2+^ toxicity by Cd^2+^-MT uptake. Rather, megalin:cubilin endocytosis of other Cd^2+^-complexed ligands, namely Cd^2+^-β2M, Cd^2+^-Alb and/or Cd^2+^-Lcn2, damages the PT.

## 4. Materials and Methods

### 4.1. Materials

Metallothionein-1 (MT) was obtained from Enzo Life Sciences, Lörrach, Germany (cat. # ALX-202-072-M001). β2-microglobulin (β2M, cat. # M4890), apo-transferrin (Tf, cat. # T2252), albumin (Alb, cat. # A6414), Chelex 100 (cat. # C7901) and 2,3-dihydroxybenzoic acid (DHBA; cat. # 126209) were obtained from Sigma-Aldrich, Taufkirchen, Germany. α1-microglobulin (α1M, cat. # ab96149) was purchased from Abcam, Berlin, Germany. Recombinant mouse NGAL/lipocalin-2 (Lcn2) (cat. # 1857-LC) was from R&D Systems, Abingdon, United Kingdom. All protein ligands had a purity of ≥95%–99% according to the suppliers, were lyophilized proteins and reconstituted in phosphate buffered saline (PBS). All other chemicals were from commercial sources and of analytical grade.

### 4.2. Methods

#### 4.2.1. Cell Culture

An immortalized cell line from the S1 segment of rat PT (WKPT-0293 Cl.2) [24] was cultured in serum-containing medium (SCM), essentially as previously described [61] with the addition of 2.5 µg/mL plasmocin (InvivoGen, Toulouse, France) in 25 cm^2^ standard tissue culture flasks (Sarstedt, Nümbrecht, Germany) at 37 °C in a humidified incubator with 5% CO_2_. Cells were passaged twice a week upon reaching confluency. Low passage cells are defined as p31-40 and high passage cells as p > 100.

#### 4.2.2. Coupling of Cd^2+^-Protein Complexes

Cd^2+^-coupled proteins (MT, Tf, Alb, β2M, α1M) were prepared exactly, as described previously [62], by mixing protein solutions with a 10-fold molar excess of CdCl_2_ solution. Briefly, MT (1 mmol/L in 10 mmol/L Tris/HCl pH 7.4) was mixed 1:1 with 10 mmol/L CdCl_2_ solution, unbound metal ions were removed with a 2:1 excess of Chelex 100 followed by centrifugation. The resulting supernatant was retained and contained 0.5 mmol/L MT complexed to 3.5 mmol/L Cd^2+^, assuming 7 Cd^2+^ ions per molecule of MT [3]. Tf, Alb, β2M or α1M (0.1 mmol/L) were complexed accordingly using a 1 mmol/L CdCl_2_ solution.

For complexation of Lcn2 with CdCl_2_, 0.5 mmol/L DHBA (in 100 mmol/L Tris/HCl, pH 8.0) and 5 mmol/L CdCl_2_ solutions were mixed 1:1 at RT for 1 h. The mixture was diluted 1:1 with 50 µmol/L Lcn2 dissolved in PBS and stirred for an additional hour. Unbound Cd^2+^ was removed with Chelex 100, as described above.

#### 4.2.3. Cell Viability Assay of WKPT-0293 Cl.2 Cells

To compensate for different cell doubling times, 12.5 × 10^3^ low-passage or 7.5 × 10^3^ high-passage WKPT-0293 Cl.2 cells were seeded per well in 24-well plates and cultured for 24 h to obtain ~30–50% confluence. Cells were then exposed to proteins or Cd^2+^-complexed proteins in modified serum-free medium (SFM_mod_; DMEM/F12, 1.2 mg/mL NaHCO_3_, 100 U/mL penicillin G, 100 µg/mL streptomycin sulfate, 4 µg/mL dexamethasone) for 24 h. After treatments, cell viability was determined by the MTT assay, as previously described [61].

#### 4.2.4. Quantitative Real-Time Polymerase Chain Reaction (qPCR)

Total RNA was isolated from confluent WKPT-0293 Cl.2 cells using High Pure RNA Isolation Kit (Roche) according to the manufacturer’s instructions and 1 μg of total RNA was reverse transcribed into cDNA using First Strand cDNA Synthesis Kit (Thermo Scientific, Schwerte, Germany). Quantitative real-time PCR was carried out using Takyon Rox SYBR MasterMix dTTP Blue (Eurogentec, Cologne, Germany) on a StepOnePlus cycler (Applied Biosystems, Schwerte, Germany). The cycling conditions were 3 min at 95 °C, followed by 40 cycles of 10 s at 95 °C and 30 s at 60 °C. Primer sequences are given in Table 2. Gapdh was used as a housekeeping gene. For analysis of relative changes, data was analyzed according to the ΔΔC_T_ method [63].

#### 4.2.5. Immunofluorescence Staining of Megalin

Low and high passage WKPT-0293 Cl.2 cells were grown on coverslips until they reached ~80% confluence. Immunofluorescence of megalin staining was performed as described previously [16]. Briefly, cells were fixed in 4% paraformaldehyde/PBS, permeabilized with 0.1% Triton X-100/PBS, blocked in 1% bovine serum albumin/PBS, incubated with primary mouse LRP2 antibody (1:200, cat. # NB110-96417, Novus Biologicals, Wiesbaden, Germany) followed by secondary donkey anti-mouse antibody conjugated to Alexa488 (1:500 cat. # 715-545-202, Dianova, Hamburg, Germany) together with the nuclear dye Hoechst 33342 (0.8 µg/mL). Coverslips were mounted onto glass slides with DAKO fluorescence mounting medium.

#### 4.2.6. Immunoblotting

SDS-PAGE and immunodetection of cubilin were performed essentially as described elsewhere [23]. For detection of β-actin, protein samples were separated by 10% SDS-PAGE. Antibodies: sheep anti-cubilin (cat. # AF3700, R&D Systems, Abingdon, United Kingdom), 1.0 µg/mL; mouse anti-β-actin (cat. # A5316, Sigma-Aldrich, Taufkirchen, Germany), 1:20,000.

#### 4.2.7. Detection of Alb and β2M Uptake

Low and high passage WKPT-0293 Cl.2 cells on coverslips (~50% confluence) were incubated with 100 nmol/L FITC-labeled Alb (cat. # A23015, Thermo Scientific, Schwerte, Germany) or 100 nmol/L β2M (cat. # ab175031, Abcam, Berlin, Germany) in SFM_mod_ for 24 h. Subsequently, cells were washed and fixed as described above. Alb treated cells were mounted without additional immunostaining. β2M exposed cells were permeabilized and stained according to the above described protocol, using a primary β2M antibody (1:500, cat. # ab175031, Abcam, Berlin, Germany) and secondary goat anti-rabbit antibody conjugated to Alexa488 (1:500, cat. # A-11008, Invitrogen, Schwerte, Germany).

#### 4.2.8. Fluorescence Imaging

Cell were imaged on a Zeiss Axiovert 200M equipped with an Fluar 40x, 1.3 NA oil immersion objective (Zeiss, Oberkochen, Germany), a CoolSnap ES camera (Roper Scientific, Planegg, Germany) and a Sola SM II light engine (Lumencor, Tübingen, Germany) using Visiview imaging software (V 3.3.0.6, Visitron Systems GmbH, Puchheim, Germany). 3–6 representative areas on each coverslip were chosen for acquisition of brightfield images and whole cell z-stacks with a step size of 800 nm for fluorescent labels. Images were analyzed and quantified using Fiji [65]. To this end, regions of interest (ROI) were drawn around each cell from which background was subtracted. The z-stack slice with the highest mean intensity for each ROI was selected for further analysis.

#### 4.2.9. Statistics

Unless otherwise indicated, the experiments were repeated at least three times with independent cultures. Bar diagrams showing means ± SEM are used for parametric data sets, unless otherwise indicated. Figures with non-parametric data are shown as box and whisker plots, presenting 25 and 75 percentiles, mean values (square symbol), median (horizontal line) and 1.5× outliers (whiskers). Scatter plots show single data points of color-coded repetitions. Statistical comparison between 2 groups was performed using Student’s unpaired t-test or Mann-Whitney-test in cases of parametric or nonparametric distribution, respectively. If more than two conditions were compared, either one-way ANOVA with Bonferroni post-hoc test (parametric) or Kruskal-Wallis with subsequent Dunn’s post-hoc test were applied using Graph-Pad Prism, San Diego, CA, USA. When one group within a specific data set showed a non-parametric distribution, the whole data set was considered as non-parametric. Results with *p* < 0.05 were considered to be statistically significant.

## Figures and Tables

**Figure 1 ijms-20-02379-f001:**
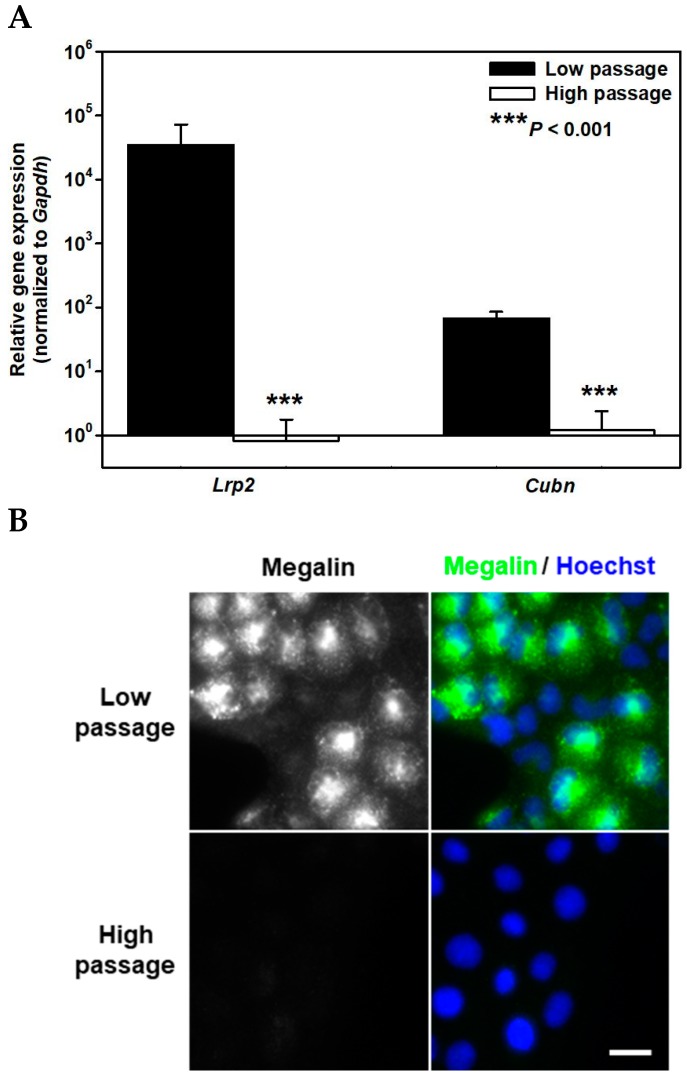
Expression of megalin and cubilin in cultured low and high passage rat renal proximal tubule cells (WKPT-0293 Cl.2). (**A**) Expression levels of megalin (*Lrp2*) and cubilin (*Cubn*) mRNA by qPCR in means ± SD of 4 experiments are shown. The C_T_ values in low passage cells were 20.6 ± 0.6 and 29.6 ± 0.1 for megalin and cubilin, respectively. Data obtained were normalized to the expression of the reference gene glyceraldehyde-3-phosphate dehydrogenase (*Gapdh*). Statistical analysis compares the low and high passage cells by unpaired *t*-test. (**B**) Expression of megalin protein in low and high passage WKPT-0293 Cl.2 cells. Megalin was detected by immunofluorescence microscopy of permeabilized cells (green). Nuclei were counterstained with Hoechst 33342 (blue). The experiment is representative of four similar ones. Scale bar = 20 µm. (**C**) Fluorescence intensity was analyzed in 188–705 cells (dots) from 4 different experiments (colors); for details of box and whisker plot statistics, see Section 4. Statistical analysis compares the low and high passage cells by non-parametric one-way ANOVA (Kruskal-Wallis-Test, using Dunn’s post-hoc analysis). (**D**) Expression of cubilin protein was detected by immunoblotting. β-actin was used as loading control. A representative experiment is shown. MM = molecular mass.

**Figure 2 ijms-20-02379-f002:**
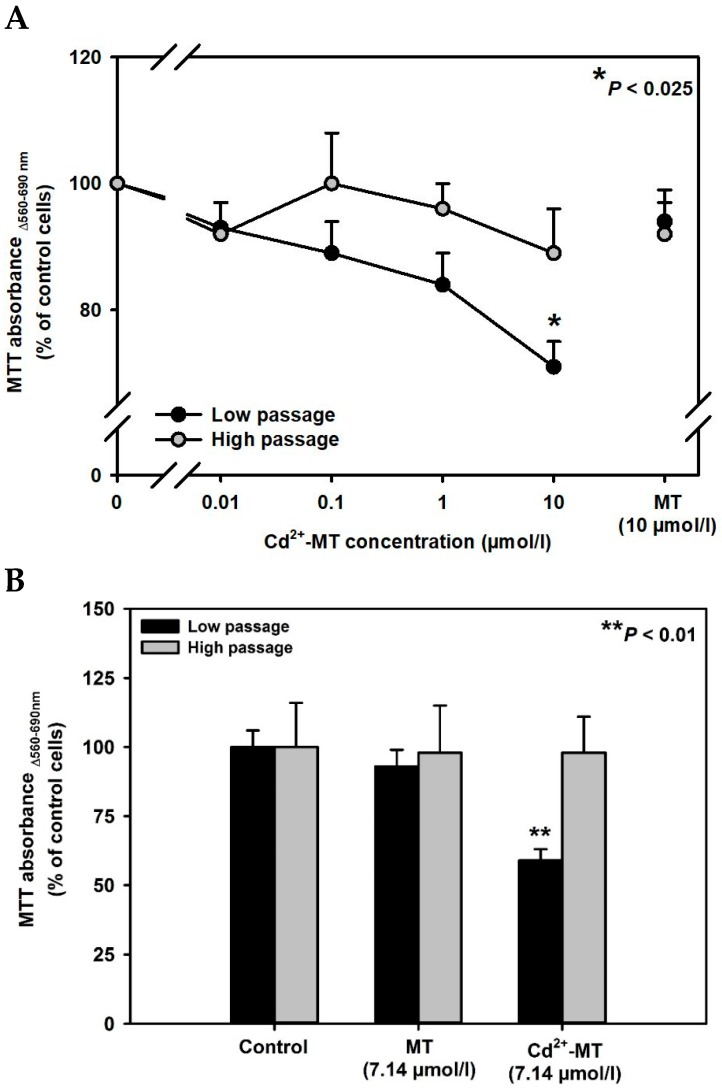
Low affinity of Cd^2+^-MT to megalin:cubilin in low passage WKPT-0293 Cl.2 cells. Treatments were performed in SFM_mod_, and cell viability was assessed using MTT assay. (**A**) Low and high passage cells were exposed to different concentrations of Cd^2+^-MT for 24 h. Means ± SEM of 8–11 experiments are shown. Absorbance values of control cells (modified serum-free medium only were 0.41 ± 0.16 a. u. (*n* = 11) and 0.51 ± 0.19 a.u. (*n* = 8)) for low and high passage cells, respectively, were set to 100%, and different Cd^2+^-MT concentrations were normalized accordingly. (**B**) Low and high passage cells were exposed to 7.14 µmol/L MT or Cd^2+^-MT (≅ 50 µmol/L for 24 h). Means ± SEM of 4–5 experiments are shown. Absorbance values of control cells were 1.06 ± 0.07 a.u. (*n* = 5) and 0.86 ± 0.14 a.u. (*n* = 4) for low and high passage cells, respectively, and set to 100%. Statistical analyses compared low to high passage cells by unpaired *t*-test.

**Figure 3 ijms-20-02379-f003:**
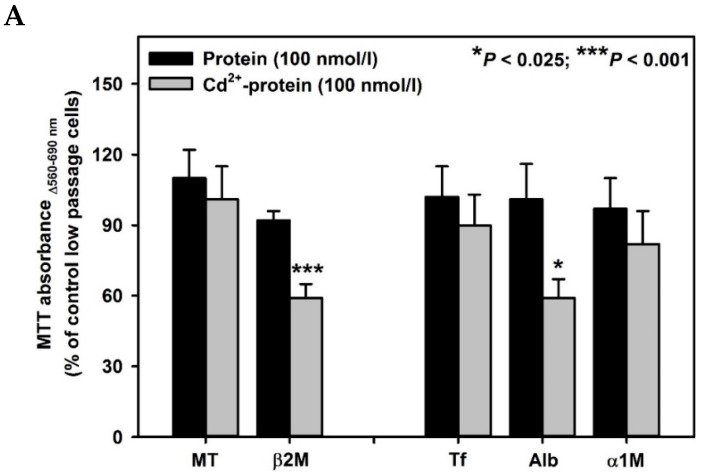
Relevant in vivo concentrations of Cd^2+^-complexes with β2-microglobulin (β2M), albumin (Alb) or lipocalin-2 (Lcn2), but not metallothionein (MT), increase toxicity in megalin:cubilin-expressing WKPT-0293 Cl.2. Treatments were performed in SFM_mod_, and cell viability was assessed using MTT assay. (**A**) Low passage cells were exposed to glomerular filtrated proteins (ligands of megalin (MT, β2M) or cubilin (Tf, Alb, α1M)) alone or complexed to Cd^2+^ for 24 h. Means ± SEM of 4–9 experiments are shown. Absorbance values of control cells were 1.06 ± 0.06 a.u. (*n* = 31) and set to 100%. (**B**) Low and high passage cells were exposed to 100 nmol/L or 1 µmol/L of the megalin ligand Lcn2 or Cd^2+^-Lcn2 for 24 h. The higher concentration of 1 µmol/L approximates in vivo concentrations of Lcn2 in the glomerular ultrafiltrate (see Table 1). Means ± SEM of 4–6 experiments are shown. Absorbance values of control cells were 1.30 ± 0.12 a.u. (*n* = 6) and 1.42 ± 0.25 a.u. (*n* = 4) for low and high passage cells, respectively, and set to 100%. Statistical analyses for pairwise comparisons were performed using unpaired *t*-test.

**Figure 4 ijms-20-02379-f004:**
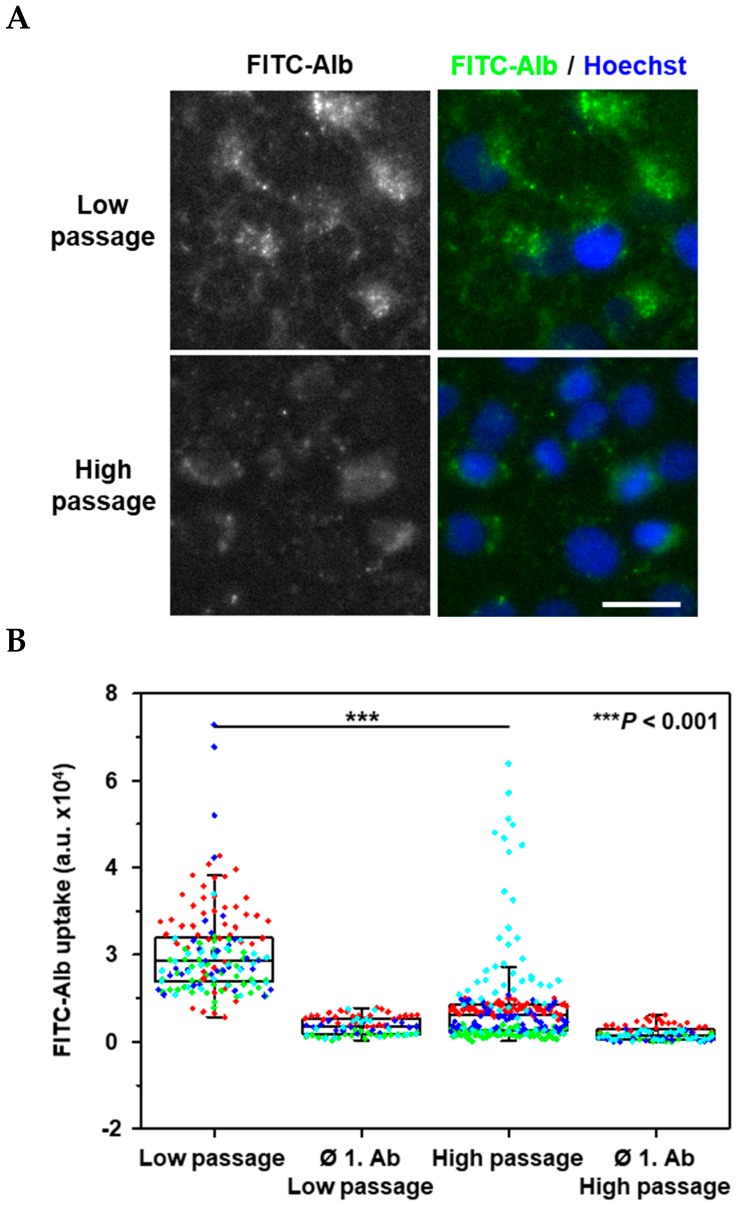
Uptake of FITC-albumin (Alb) in low and high passage WKPT-0293 Cl.2 cells. (**A**) Cells were exposed to 100 nmol/L FITC-Alb (green) in SFM_mod_ for 24 h. Subsequently, cells were fixed, and mounted for fluorescence imaging. Nuclei were counterstained with Hoechst 33342 (blue). The experiment is representative of four similar ones. Scale bar = 20 µm. (**B**) Fluorescence intensity was analyzed in 83–238 cells (dots) from 4 different experiments (colors). For details of box plot statistics, see Section 4.2. Statistical analysis compared the low and high passage cells by non-parametric one-way ANOVA (Kruskal-Wallis-Test, using Dunn’s post-hoc analysis.

**Table 1 ijms-20-02379-t001:** Binding affinities and estimated concentrations of ligands of megalin and cubilin in the renal glomerular filtrate.

Ligand	Receptor	*K_D_* (nmol/L)	Reference	Concentration in Glomerular Filtrate (nmol/L) *
**MT**	Megalin	100,000	[22]	0.5–5
**β2M**	Megalin	420	[30]	100
**Lcn2**	Megalin	60	[31]	650
**Tf**	Cubilin	20	[27]	2
**Alb**	Cubilin	630	[28]	53
**α1M**	Cubilin	n.d.	Ø	92

* Calculations are based on estimated glomerular sieving coefficients of plasma proteins [32].

**Table 2 ijms-20-02379-t002:** qPCR primers.

Gene	Forward Primer (5′-3′)	Reverse Primer (5′-3′)
Rat megalin/*Lrp2* (NM_030827.1) ^a^	TGGAATCTCCCTTGATCCTG	TGTTGCTGCCATCAGTCTTC
Rat cubilin/*Cubn* (NM_053332) ^a^	GCACTGGCAATGAACTAGCA	TGATCCAGGAGCACTCTGTG
Rat *Gapdh* (NM_017008)	AGGGCTCATGACCACAGT	TGCAGGGATGATGTTCTG

^a^ Primer sequences taken from Prabakaran et al. [64].

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
