# Peer review of "Cadmium Complexed with β2-Microglubulin, Albumin and Lipocalin-2 rather than Metallothionein Cause Megalin:Cubilin Dependent Toxicity of the Renal Proximal Tubule"

_ijms, 2019, doi:10.3390/ijms20102379_

Round 1

Reviewer 1 Report

General Comments:

The authors wanted to show in this manuscript that Cd2+-MT is less toxic than Cd2+-b2M, Cd2+-Alb and Cd2+-Lcn2. Based on their findings they want to argue the current dogma that Cd2+-MT endocytosed via megalin:cubilin causing renal damage. However, the manuscript represented limited data and poor quality imaging methods that need to be address. I have few concerns with the author's design and presentation of their data as mentioned below.

Specific Comments:

Abstract - 

1. Line 16The authors introduced Cd2+ as an environmental threat but there is no case or cities specific information. They discuss more about food and smoking related exposure. Can the authors elaborate?

2. Line 24: The study was based on a rat model and yet to be confirmed in human samples. So I feel that the authors overstated their claim and disregard previous findings. Any reasons?

3. Line 35-36 : I agree that they would be good candidate to investigate. 

Introduction - 

1. Line 61, 72,79 etc: Why does the authors add comments to their references? I found it throughout the manuscript. 

2. Line 85 : What type of cells is WKPT-0293 Cl.2? hepatocytes or kupffer cells? Line 52,53

Methods - 

1. Line 93-98: All the ligands/ proteins were from different source. Did the authors check the purity? Did the authors reconstitute them in the same solvent?

2. Line 103: What is the function of plasmocin?

3. Line 106-117: Did the authors perform any biophysical assay like ITC, to check the protein-Cd2+ binding?

4. Line 119: Why did the authors used different cell number for MTT assay? 

5. Line 120, 147: Please correct to "cells were stimulated with the proteins" instead of expose.

6. Line 134-142: During IF staining, did the authors saw died/lost of cells due to toxicity? 

7. Line 152: The authors used a 40X oil immersion lens, is that correct or typo? I thought oil immersion were available for 60X and above. If yes, why didn't the authors use a high magnification objective considering that the data is the most crucial for their claim?

Results - 

1. Line 177: IF staining was done with anti-human? It is confusing since its mentioned as cross react in method section.

2. Figure 1: Panel A and B could have been one panel. It looks that megalin have high expression compared to cubilin. Does the authors have any reason? Panel B, image is very poor. Did the authors thought of doing a western blot instead? Panel C, Why do we see a tail in the low passage samples? The caption does not explain about the blue and green color in panel C. 

Did the authors check the effect of Cd2+ alone as a control?

3. Figure 2: Panel A is a proof of concept since other studies have shown the pattern of toxicity by MT. Why there is toxic effect at 0umol/l conc. for both low and high passage? Panel B, why did the authors selected 7.14umol/l? 

4. Figure 3: Panel A, MT bar does not match with Fig 2A shows at 100nmol/l or 0.1umol/l, it was ~90% in Fig 2A but it is higher in Fig 3A, MT. Does the authors have any reason? Why did the authors use different conc. units? It will confuse the readers.

5. Why didn't the authors used an alternative technique to confirm toxicity? Ex, annexin/PI staining.

6. Fig 4A: Poor quality imaging. 4B: Why do the authors find a tail in high passage samples? The authors need to design a neat experiments to achieve the uptake efficiency. 

7. In general, did the authors thought about the different capacity of kupffer versus hepatocytes for protein uptake if there exist a megalin:cubilin independent uptake? 

References - 

1. The WHO report is eight years old. The authors did not find a recent report or the public health concern with cadmium is neglected now?

2. The authors did a good review of literature.

Author Response

Reviewer 1

We thank the expert reviewer for her/his comments and suggestions, which have helped to improve the manuscript.

General Comments:

The authors wanted to show in this manuscript that Cd2+-MT is less toxic than Cd2+-b2M, Cd2+-Alb and Cd2+-Lcn2. Based on their findings they want to argue the current dogma that Cd2+-MT endocytosed via megalin:cubilin causing renal damage. However, the manuscript represented limited data and poor quality imaging methods that need to be address. I have few concerns with the author's design and presentation of their data as mentioned below.

Specific Comments:

Abstract - 

1. Line 16The authors introduced Cd2+ as an environmental threat but there is no case or cities specific information. They discuss more about food and smoking related exposure. Can the authors elaborate? The abstract is very limited on word number (ca. 200 words) therefore it is does not allow one to elaborate on this issue. Moreover, Cd remains in the environment because Cd cannot be degraded further hence the location of Cd contamination is irrelevant with respect to impact on environmental pollution. Reference 1 is a complete assessment of environmental Cd toxicity and the associated health risks and gives the reader ample information on this issue.

2. Line 24: The study was based on a rat model and yet to be confirmed in human samples. So I feel that the authors overstated their claim and disregard previous findings. Any reasons?  Previous findings regarding handling of Cd complexes by the kidney have not been disregarded by us. The current dogma derives from animal and cell culture experiments, in fact, mostly from rat models. Moreover, human data are indirect. These data are urinary or renal function data and correlated with experimental cell and animal experiment data. In contrast to the inaccurate current dogma, the conclusions made in our manuscript take into account careful consideration and calculations for glomerular ligand filtration, ligand-receptor binding affinities and availability of ligands in the kidney. In fact, the rat cell line model is the most appropriate model as a direct comparison to previous findings, which were executed in the rat. To our knowledge, no human experiments are available. Such experiments would need human kidney organoids, which have only been developed recently.

3. Line 35-36 : I agree that they would be good candidate to investigate.  O.k.

Introduction - 

1. Line 61, 72,79 etc: Why does the authors add comments to their references? I found it throughout the manuscript. These are not comments, but indicate a review article is being cited rather than an original research paper.

2. Line 85 : What type of cells is WKPT-0293 Cl.2? hepatocytes or kupffer cells? Line 52,53 As stated in the title and throughout the manuscript, this paper deals with the kidney, not the liver. As detailed in the Introduction, Cd is thought to accumulate at first in the liver and then to redistribute to the kidney where it accumulates and causes kidney damage. The cell line is a kidney proximal tubule cell line derived from Wistar Kyoto rats and is clearly stated in the Methods section (subheading Cell Culture).

Methods - 

1. Line 93-98: All the ligands/ proteins were from different source. Did the authors check the purity? Did the authors reconstitute them in the same solvent? The proteins all had a purity of ≥95 - 99% according to the suppliers, were lyophilized proteins and reconstituted in PBS. A sentence in the text has been included on page 3, line 100.

2. Line 103: What is the function of plasmocin? Plasmocin prevents growth of mycoplasma, which is common with cultured cells. This is a standard procedure for cell culture and does not need explanation in the text.

3. Line 106-117: Did the authors perform any biophysical assay like ITC, to check the protein-Cd2+ binding? Protein-Cd binding has already been performed by other authors (see references 3, 18-20) thus relinquishing the necessity to execute these experiments.

4. Line 119: Why did the authors used different cell number for MTT assay? High passage cells proliferate faster than low passage cells. The duplication time of low passage cells is about 25 hours and of high passage cells about 20 hours hence different cell numbers are necessary to obtain similar confluence prior to Cd exposure. A sentence in the text explains this fact (line 123). Please, note that during treatment cells are cultured in serum-free medium and without growth factors (insulin, EGF) that are all ligands of megalin, and therefore do not divide anymore.

5. Line 120, 147: Please correct to "cells were stimulated with the proteins" instead of expose. We disagree with the reviewer. The cells were not stimulated because the proteins were endocytosed and degraded in lysosomes. The term “expose” reflects the fact that the proteins are included in the extracellular medium.

6. Line 134-142: During IF staining, did the authors saw died/lost of cells due to toxicity?  Lines 141-147 (revised manuscript) refer to IF staining in the absence of Cd-protein complexes. Hence, no toxicity occurs. This staining describes the expression of megalin in control cells.

7. Line 152: The authors used a 40X oil immersion lens, is that correct or typo? I thought oil immersion were available for 60X and above. If yes, why didn't the authors use a high magnification objective considering that the data is the most crucial for their claim? No, this is not a typo. We used a Fluar x40, 1.3 oil-immersion objective from Zeiss (see https://www.micro-shop.zeiss.com/en/us/shop/objectives"). In contrast to the reviewer, we feel a higher magnification would be only beneficial for qualitative aspects of megalin expression and protein ligand uptake (e.g. subcellular distribution, co-localization with certain organelles, etc.), whereas the questions addressed here relate to quantitative aspects.

Results - 

1. Line 177: IF staining was done with anti-human? It is confusing since its mentioned as cross react in method section. These proteins are highly conserved across species (check the data sheet https://www.novusbio.com/products/lrp2-antibody-cd7d5_nb110-96417). Human and rat megalin are almost 80% identical at the amino acid level. For the sake of clarity, we have removed the word “human”.

2. Figure 1: Panel A and B could have been one panel. It looks that megalin have high expression compared to cubilin. Does the authors have any reason? Panel B, image is very poor. Did the authors thought of doing a western blot instead? Panel C, Why do we see a tail in the low passage samples? The caption does not explain about the blue and green color in panel C.  The question why megalin is more expressed than cubilin is a good question. This has not been clarified so far. Yet, this is also observed in vivo in rat proximal tubule (see ref. 21). A sentence has been added to acknowledge this fact. Images have been replaced with better quality images. In the legend to panel B (the reviewer likely means panel B and not C), blue and green colors have been defined. A western blot of cubilin in low and high passage cells has been performed (see lines 148-152, 195-196 and Figure 1D). A western blot of megalin has not been performed because the protein has a MM of ~600 kDa and requires expertise that we would need to develop in the laboratory. The caption clearly explains the meaning of the different colors: 4 different colors were used (including blue and green) to show 4 different experiments. Individual points show staining intensity in individual cells. Tail: Individual cells within a population of cells in a cell line differ slightly in their genotype, and even more in their phenotype. Hence, the measured single cell intensities will differ. This feature is more than useful, as it reflects natural variations in a specific nephron segment. Additionally, expression of certain proteins (e.g. in our case megalin) will vary from cell batch to batch, even if the average expression level within a population (high or low passage cells) does not differ significantly. The style of data presentation is transparent (we do not want to hide natural variations by simply showing a box plot without a combined scatter plot).

Did the authors check the effect of Cd2+ alone as a control? The suggested experiments have been previously published using the low passage cells (see for example experiments in Nair AR et al. Arch Toxicol. 2015 Dec;89(12):2273-89). Yet no comparison has been made between low and high passage cells. These experiments have been included in the revised manuscript using 700 nmol/l CdCl2 and 50 µmol/l CdCl2 (corresponding to 100 nmol/l MT and 7.14 µmol/l MT because MT has 7 Cd binding sites). As expected, 700 nmol/l CdCl2 showed no toxicity in low and high passage cells, whereas 50 µmol/l CdCl2 decreased cell viability to 0% in both, low and high passage cells (n = 4). This indicates that the mechanisms of CdCl2 toxicity differ from Cd-MT toxicity and also prove that Cd is completely chelated by MT.This observation has been included in the revised manuscript (lines 228-231).

3. Figure 2: Panel A is a proof of concept since other studies have shown the pattern of toxicity by MT. Why there is toxic effect at 0umol/l conc. for both low and high passage? Panel B, why did the authors selected 7.14umol/l?  This is not a proof of concept that Cd-MT is toxic. It proves that low passage cells (that express megalin) show toxicity whereas high passage cells (that do not express megalin) do not show CdMT toxicity. Moreover, the experiment shows that toxicity only occurs at Cd-coupled MT concentrations of 10 µmol/l or higher, which confirms that megalin has a low affinity to MT. All this reasoning was already present in the text of the manuscript. The text of the manuscript + the legend also indicates that 0 µmol/l Cd-MT corresponds to 10 µmol/l apo-MT, which has a slight toxicity in both cell lines. 100 % is the MTT absorbance of control cells without MT or Cd-MT. To avoid confusion, Figure 2A has been changed such that 10 µmol/l apo-MT is displayed as a separate point. In panel B, we select 7.14 µmol/l MT because MT binds 7 Cd ions, therefore the concentration of Cd is 50 µmol/l. This concentration has previously been used in various publications from my group, including references 23 and 26, and showed robust megalin-dependent toxic effects of Cd-MT. This reasoning was already present in the text of the previous version of the manuscript.

4. Figure 3: Panel A, MT bar does not match with Fig 2A shows at 100nmol/l or 0.1umol/l, it was ~90% in Fig 2A but it is higher in Fig 3A, MT. Does the authors have any reason? Why did the authors use different conc. units? It will confuse the readers. The mean values are slightly different (90% versus 101%). However when the errors are taken into account, it becomes obvious that the differences are normal scatter (90 ± 5 % in Figure 2A and 101 ± 14% in Figure 3A). But this point is irrelevant because, both in Figures 2A and 3A, 100 nmol/l Cd-MT does NOT show significant toxicity (toxicity becomes significant at 7-10 µmol/l for low passage cells only)! The reason for using 100 nmol/l is explained in the text of the original manuscript and hs been improved in the revision (lines 249-270). Several putative ligands of megalin have a Kd for megalin binding at around 100 nmol/l and the concentrations found in the primary filtrate are about 100 nmol/l. Therefore, all the tested Cd-protein ligands were investigated at 100 nmol/l to compare the relative toxicity.

5. Why didn't the authors used an alternative technique to confirm toxicity? Ex, annexin/PI staining. In principle, another method may be used. However, CdCl2/Cd-MT toxicity by MTT assay nicely correlates with other apoptosis/cell death assays in these cells (e.g. Annexin-V, Hoechst 33342, cleaved caspase 3, etc.). We have extensively characterized these various techniques in our previous studies (e. g. ref. 25; Lee WK, Abouhamed M, Thévenod F. Am J Physiol Renal Physiol. 2006 Oct;291(4):F823-32).

6. Fig 4A: Poor quality imaging. 4B: Why do the authors find a tail in high passage samples? The authors need to design a neat experiments to achieve the uptake efficiency. The images have been replaced with high quality images. Tail: Regarding the issue on the scattering of the individual cell signals: please see comment on similar concern to Figure 1C. The experiments show 1) biological variability and 2) indicate that other processes are involved that contribute to uptake of proteins independently from toxicity (see lines 344-347 in the discussion). They are the aim of current studies that will be published in the future because they are beyond the scope of this study.

7. In general, did the authors thought about the different capacity of kupffer versus hepatocytes for protein uptake if there exist a megalin:cubilin independent uptake? As stated in the title and throughout the manuscript, this paper deals with the kidney, not the liver. Cd is thought to accumulate at first in the liver and then to redistribute to the kidney where it accumulates and causes kidney damage. The cell line is a kidney proximal tubule cell line. The question asked by the reviewer is indeed interesting, but not the aim of this study.

References - 

1. The WHO report is eight years old. The authors did not find a recent report or the public health concern with cadmium is neglected now? As stated before, Cd in the environment remains in the environment because Cd cannot be degraded further. Reference 1 is the most authoritative to emphasize the damage caused by Cd in the environment.

2. The authors did a good review of literature. Thank you.

Reviewer 2 Report

Relevant study affording the unresolved problem of cadmium toxicity. Suggests that previous literature should be carefully reconsidered.

Manuscript clearly written, original and worth pubblication.

minor requests:

1) please no acronims in abstract. Move in introduction.

2) line 133, table 1 heading: please specify the author name before number.

3) line 160: All replicates are biological replicates. please explain. 

4) figures. increase in size X and Y titles, numbering etc since they are hardly visible.

5) table 2: too dispersive. Please compact

Author Response

Reviewer 2

We thank the expert reviewer for her/his constructive suggestions, which have helped to improve the manuscript.

Relevant study affording the unresolved problem of cadmium toxicity. Suggests that previous literature should be carefully reconsidered.

Manuscript clearly written, original and worth publication. Thank you!

minor requests:

1) please no acronyms in abstract. Move in introduction. O.K., changed as suggested.

2) line 133, table 1 heading: please specify the author name before number. O.K., changed as suggested.

3) line 160: All replicates are biological replicates. please explainBiological replicates = multiple samples/batches of cells; technical replicates = test the same sample multiple times to determine variability. To avoid confusion, we removed the sentence.

4) figures. increase in size X and Y titles, numbering etc since they are hardly visible. O.k., changed as suggested. Images now portrait rather than landscape.

5) table 2: too dispersive. Please compact. The format was determined by the journal. O.K. Changed, as suggested.

Round 2

Reviewer 1 Report

General Comments:

The authors wanted to show in this manuscript that Cd2+-MT is less toxic than Cd2+-b2M, Cd2+-Alb and Cd2+-Lcn2. Based on their findings they want to argue the current dogma that Cd2+-MT endocytosed via megalin:cubilin causing renal damage. However, the manuscript represented limited data and poor quality imaging methods that need to be address. I have few concerns with the author's design and presentation of their data as mentioned below.

Specific Comments:

Abstract -

1. Line 16The authors introduced Cd2+ as an environmental threat but there is no case or cities specific information. They discuss more about food and smoking related exposure. Can the authors elaborate? The abstract is very limited on word number (ca. 200 words) therefore it is does not allow one to elaborate on this issue. Moreover, Cd remains in the environment because Cd cannot be degraded further hence the location of Cd contamination is irrelevant with respect to impact on environmental pollution. Reference 1 is a complete assessment of environmental Cd toxicity and the associated health risks and gives the reader ample information on this issue.

Reviewer: I am aware about the word limit in abstract section. I meant to elaborate in the background/ Introduction section. The word count for the revised manuscript: Abstract is ~300 words.

2. Line 24: The study was based on a rat model and yet to be confirmed in human samples. So I feel that the authors overstated their claim and disregard previous findings. Any reasons? Previous findings regarding handling of Cd complexes by the kidney have not been disregarded by us. The current dogma derives from animal and cell culture experiments, in fact, mostly from rat models. Moreover, human data are indirect. These data are urinary or renal function data and correlated with experimental cell and animal experiment data. In contrast to the inaccurate current dogma, the conclusions made in our manuscript take into account careful consideration and calculations for glomerular ligand filtration, ligand-receptor binding affinities and availability of ligands in the kidney. In fact, the rat cell line model is the most appropriate model as a direct comparison to previous findings, which were executed in the rat. To our knowledge, no human experiments are available. Such experiments would need human kidney organoids, which have only been developed recently.

Reviewer: Can understand limitations.

3. Line 35-36 : I agree that they would be good candidate to investigate. O.k.

Introduction -

1. Line 61, 72,79 etc: Why does the authors add comments to their references? I found it throughout the manuscript. These are not comments, but indicate a review article is being cited rather than an original research paper.

Reviewer: It is very unusual to see such highlights. Readers can figure it out if it is an article or review. If the authors want to emphasize them as an important review I recommend adding them in the reference section.

2. Line 85 : What type of cells is WKPT-0293 Cl.2? hepatocytes or kupffer cells? Line 52,53 As stated in the title and throughout the manuscript, this paper deals with the kidney, not the liver. As detailed in the Introduction, Cd is thought to accumulate at first in the liver and then to redistribute to the kidney where it accumulates and causes kidney damage. The cell line is a kidney proximal tubule cell line derived from Wistar Kyoto rats and is clearly stated in the Methods section (subheading Cell Culture).

Reviewer: Thanks for your explanation.  

Methods -

1. Line 93-98: All the ligands/ proteins were from different source. Did the authors check the purity? Did the authors reconstitute them in the same solvent? The proteins all had a purity of ≥95 - 99% according to the suppliers, were lyophilized proteins and reconstituted in PBS. A sentence in the text has been included on page 3, line 100.

Reviewer: Good. Checked it.

2. Line 103: What is the function of plasmocin? Plasmocin prevents growth of mycoplasma, which is common with cultured cells. This is a standard procedure for cell culture and does not need explanation in the text.

Reviewer: This concern was raised to consider readers from different fields. The second sentence could have been avoided.

3. Line 106-117: Did the authors perform any biophysical assay like ITC, to check the protein-Cd2+ binding? Protein-Cd binding has already been performed by other authors (see references 3, 18-20) thus relinquishing the necessity to execute these experiments.

Reviewer: So, there was no problem with any of the assay during the experiments. Good.

4. Line 119: Why did the authors used different cell number for MTT assay? High passage cells proliferate faster than low passage cells. The duplication time of low passage cells is about 25 hours and of high passage cells about 20 hours hence different cell numbers are necessary to obtain similar confluence prior to Cd exposure. A sentence in the text explains this fact (line 123). Please, note that during treatment cells are cultured in serum-free medium and without growth factors (insulin, EGF) that are all ligands of megalin, and therefore do not divide anymore.

Reviewer: Good correction and explanation.

5. Line 120, 147: Please correct to "cells were stimulated with the proteins" instead of expose. We disagree with the reviewer. The cells were not stimulated because the proteins were endocytosed and degraded in lysosomes. The term “expose” reflects the fact that the proteins are included in the extracellular medium.

Reviewer: I am confused with the authors comment. There are many cell immunology literatures which will support my suggestion. They could have just mentioned “added” then.

6. Line 134-142: During IF staining, did the authors saw died/lost of cells due to toxicity? Lines 141-147 (revised manuscript) refer to IF staining in the absence of Cd-protein complexes. Hence, no toxicity occurs. This staining describes the expression of megalin in control cells.

Reviewer: Good revision.

7. Line 152: The authors used a 40X oil immersion lens, is that correct or typo? I thought oil immersion were available for 60X and above. If yes, why didn't the authors use a high magnification objective considering that the data is the most crucial for their claim? No, this is not a typo. We used a Fluar x40, 1.3 oil-immersion objective from Zeiss (see https://www.micro-shop.zeiss.com/en/us/shop/objectives"). In contrast to the reviewer, we feel a higher magnification would be only beneficial for qualitative aspects of megalin expression and protein ligand uptake (e.g. subcellular distribution, co-localization with certain organelles, etc.), whereas the questions addressed here relate to quantitative aspects.

Reviewer: Ok.

Results -

1. Line 177: IF staining was done with anti-human? It is confusing since its mentioned as cross react in method section. These proteins are highly conserved across species (check the data sheet https://www.novusbio.com/products/lrp2-antibody-cd7d5_nb110-96417). Human and rat megalin are almost 80% identical at the amino acid level. For the sake of clarity, we have removed the word “human”.

Reviewer: Good for the readers.

2. Figure 1: Panel A and B could have been one panel. It looks that megalin have high expression compared to cubilin. Does the authors have any reason? Panel B, image is very poor. Did the authors thought of doing a western blot instead? Panel C, Why do we see a tail in the low passage samples? The caption does not explain about the blue and green color in panel C. The question why megalin is more expressed than cubilin is a good question. This has not been clarified so far. Yet, this is also observed in vivo in rat proximal tubule (see ref. 21). A sentence has been added to acknowledge this fact. Images have been replaced with better quality images. In the legend to panel B (the reviewer likely means panel B and not C), blue and green colors have been defined. A western blot of cubilin in low and high passage cells has been performed (see lines 148-152, 195-196 and Figure 1D). A western blot of megalin has not been performed because the protein has a MM of ~600 kDa and requires expertise that we would need to develop in the laboratory. The caption clearly explains the meaning of the different colors: 4 different colors were used (including blue and green) to show 4 different experiments. Individual points show staining intensity in individual cells. Tail: Individual cells within a population of cells in a cell line differ slightly in their genotype, and even more in their phenotype. Hence, the measured single cell intensities will differ. This feature is more than useful, as it reflects natural variations in a specific nephron segment. Additionally, expression of certain proteins (e.g. in our case megalin) will vary from cell batch to batch, even if the average expression level within a population (high or low passage cells) does not differ significantly. The style of data presentation is transparent (we do not want to hide natural variations by simply showing a box plot without a combined scatter plot).

Reviewer: Appreciate the authors for correction and doing an additional experiment. The revision looks better and conclusive now.

Did the authors check the effect of Cd2+ alone as a control? The suggested experiments have been previously published using the low passage cells (see for example experiments in Nair AR et al. Arch Toxicol. 2015 Dec;89(12):2273-89). Yet no comparison has been made between low and high passage cells. These experiments have been included in the revised manuscript using 700 nmol/l CdCl2 and 50 µmol/l CdCl2 (corresponding to 100 nmol/l MT and 7.14 µmol/l MT because MT has 7 Cd binding sites). As expected, 700 nmol/l CdCl2 showed no toxicity in low and high passage cells, whereas 50 µmol/l CdCl2 decreased cell viability to 0% in both, low and high passage cells (n = 4). This indicates that the mechanisms of CdCl2 toxicity differ from Cd-MT toxicity and also prove that Cd is completely chelated by MT.This observation has been included in the revised manuscript (lines 228-231).

Reviewer: Thanks for the explanation.

3. Figure 2: Panel A is a proof of concept since other studies have shown the pattern of toxicity by MT. Why there is toxic effect at 0umol/l conc. for both low and high passage? Panel B, why did the authors selected 7.14umol/l? This is not a proof of concept that Cd-MT is toxic. It proves that low passage cells (that express megalin) show toxicity whereas high passage cells (that do not express megalin) do not show CdMT toxicity. Moreover, the experiment shows that toxicity only occurs at Cd-coupled MT concentrations of 10 µmol/l or higher, which confirms that megalin has a low affinity to MT. All this reasoning was already present in the text of the manuscript. The text of the manuscript + the legend also indicates that 0 µmol/l Cd-MT corresponds to 10 µmol/l apo-MT, which has a slight toxicity in both cell lines. 100 % is the MTT absorbance of control cells without MT or Cd-MT. To avoid confusion, Figure 2A has been changed such that 10 µmol/l apo-MT is displayed as a separate pointIn panel B, we select 7.14 µmol/l MT because MT binds 7 Cd ions, therefore the concentration of Cd is 50 µmol/l. This concentration has previously been used in various publications from my group, including references 23 and 26, and showed robust megalin-dependent toxic effects of Cd-MT. This reasoning was already present in the text of the previous version of the manuscript.

Reviewer: Ok. Checked references.

4. Figure 3: Panel A, MT bar does not match with Fig 2A shows at 100nmol/l or 0.1umol/l, it was ~90% in Fig 2A but it is higher in Fig 3A, MT. Does the authors have any reason? Why did the authors use different conc. units? It will confuse the readers. The mean values are slightly different (90% versus 101%). However when the errors are taken into account, it becomes obvious that the differences are normal scatter (90 ± 5 % in Figure 2A and 101 ± 14% in Figure 3A). But this point is irrelevant because, both in Figures 2A and 3A, 100 nmol/l Cd-MT does NOT show significant toxicity (toxicity becomes significant at 7-10 µmol/l for low passage cells only)! The reason for using 100 nmol/l is explained in the text of the original manuscript and hs been improved in the revision (lines 249-270). Several putative ligands of megalin have a Kd for megalin binding at around 100 nmol/l and the concentrations found in the primary filtrate are about 100 nmol/l. Therefore, all the tested Cd-protein ligands were investigated at 100 nmol/l to compare the relative toxicity.

Reviewer: Interesting to know.

5. Why didn't the authors used an alternative technique to confirm toxicity? Ex, annexin/PI staining. In principle, another method may be used. However, CdCl2/Cd-MT toxicity by MTT assay nicely correlates with other apoptosis/cell death assays in these cells (e.g. Annexin-V, Hoechst 33342, cleaved caspase 3, etc.). We have extensively characterized these various techniques in our previous studies (e. g. ref. 25; Lee WK, Abouhamed M, Thévenod F. Am J Physiol Renal Physiol. 2006 Oct;291(4):F823-32).

Reviewer: Every new study need confirmation. So, my concern was to get a similar pattern using different technique.

6. Fig 4A: Poor quality imaging. 4B: Why do the authors find a tail in high passage samples? The authors need to design a neat experiments to achieve the uptake efficiency. The images have been replaced with high quality images. Tail: Regarding the issue on the scattering of the individual cell signals: please see comment on similar concern to Figure 1C. The experiments show 1) biological variability and 2) indicate that other processes are involved that contribute to uptake of proteins independently from toxicity (see lines 344-347 in the discussion). They are the aim of current studies that will be published in the future because they are beyond the scope of this study.

Reviewer: Good to know that the authors will address in their next study.

7. In general, did the authors thought about the different capacity of kupffer versus hepatocytes for protein uptake if there exist a megalin:cubilin independent uptake? As stated in the title and throughout the manuscript, this paper deals with the kidney, not the liver. Cd is thought to accumulate at first in the liver and then to redistribute to the kidney where it accumulates and causes kidney damage. The cell line is a kidney proximal tubule cell line. The question asked by the reviewer is indeed interesting, but not the aim of this study.

Reviewer: Ok.

References -

1. The WHO report is eight years old. The authors did not find a recent report or the public health concern with cadmium is neglected now? As stated beforeCd in the environment remains in the environment because Cd cannot be degraded further. Reference 1 is the most authoritative to emphasize the damage caused by Cd in the environment.

Reviewer: Interesting!

2. The authors did a good review of literature. Thank you.